# In Vitro Detoxification of Fumonisin B1 (FB1) into Hydrolyzed Fumonisin B1 (HFB1) by *Lactobacillus* spp. Isolated from Pig Caecum

**DOI:** 10.3390/ijms262110557

**Published:** 2025-10-30

**Authors:** Huu Anh Dang, Attila Zsolnai, Mariam Kachlek, Veronika Halas, Diana Giannuzzi, Stefano Schiavon, Isaac Hyeladi Malgwi

**Affiliations:** 1Department of Veterinary Microbiology and Infectious Diseases, Faculty of Veterinary Medicine, Vietnam National University of Agriculture, Gia Lam, Hanoi 12400, Vietnam; dhanh@vnua.edu.vn; 2Institute of Animal Husbandry Sciences, Hungarian University of Agriculture and Life Sciences, Magyar Agrár- és Élettudományi Egyetem (MATE), Kaposvár Campus, Guba Sándor utca 40., H-7400 Kaposvár, Hungary; 3Animal Management and Saddlery School—Enfield Campus, Capel Manor College, Bullsmoor Lane, Enfield EN1 4RQ, UK; mariam.kachlek@capel.ac.uk; 4Agribiotechnology and Precision Breeding for Food Security National Laboratory, Department of Farm Animal Nutrition, Institute of Physiology and Nutrition, Hungarian University of Agriculture and Life Sciences, Magyar Agrár- és Élettudományi Egyetem (MATE), Kaposvár Campus, Guba Sándor utca 40., H-7400 Kaposvár, Hungary; halas.veronika@uni-mate.hu; 5Department of Agronomy, Food, Natural Resources, Animals and Environment (DAFNAE), University of Padova, Capmus Agripolis, Viale dell’Università 16, I-35020 Legnaro, Italy; diana.giannuzzi@unipd.it (D.G.); stefano.schiavon@unipd.it (S.S.)

**Keywords:** feed additives, gut toxicity, hydrolysis, probiotics, swine, intestinal microbiota

## Abstract

The metabolic capacity of swine caecum-derived *Lactobacillus* spp. to biotransform mycotoxins presents promising potential as a host-probiotic strategy to improve pig health and support host-targeted probiotic research. In the present study, *Lactobacillus* spp. isolated from the pig caecum were examined for their ability to detoxify fumonisin B1 (FB1) in vitro. Three experimental groups were established (i) Control 1 (C1: buffer + caecal chyme), (ii) Control 2 (C2: buffer + FB1), (iii) Experimental group (E: buffer + caecal chyme + FB1), each with 12 replicates per group (4 replicates per time point 0, 24, and 48 h). Quantitative polymerase chain reaction (qPCR) was used to determine bacterial abundance, while fumonisin B1 (FB1) and its hydrolyzed product (HFB1, Hydrolyzed Fumonisin B1), were quantified using liquid chromatography–mass spectrometry (LC-MS). Group E showed a significant increase in *Lactobacillus* spp. abundance (*p* < 0.001), indicating a selective microbial response to FB1 exposure. In contrast, total bacterial counts did not differ significantly between C1 and E (*p* = 0.35), suggesting that the proliferation of *Lactobacillus* was the main microbiological outcome supporting the host–probiotic hypothesis. Principal component analysis (PCA) revealed distinct microbial clustering, explaining 97.3% of the variance. Compared to C2, FB1 levels in group E were significantly reduced at 24 and 48 h, while HFB1 conversion rates increased from 47.1% to 56.5%. The study identified *Lactobacillus pontis*, *Lactobacillus amylovorus*, and *Lactobacillus ultunensis* as promising host-associated probiotics, with potential application as feed additives to mitigate mycotoxin effects in pigs. These findings warrant further in vivo validation.

## 1. Introduction

Mycotoxins are toxic secondary metabolites produced by certain fungi. Some mycotoxins, including deoxynivalenol, zearalenone, fumonisins, aflatoxins, and ochratoxins, can adsorb onto bacterial cell walls, and this interaction is influenced by physicochemical factors such as pH, temperature, and cell wall composition [1,2,3]. Fumonisins, a class of water-soluble mycotoxins (e.g., fumonisin B1 and B2), are primarily produced by Fusarium verticillioides and Fusarium proliferatum and commonly contaminate maize and maize-based products [4,5]. They are classified as possibly carcinogenic to humans (IARC Group 2B) [6,7], disrupt sphingolipid metabolism, and cause porcine pulmonary oedema (PPE) in pigs, equine leukoencephalomalacia (ELEM) in horses, and hepatic and renal damage in various other animal species [8,9]. Consequently, the European Union has established guidance levels for fumonisin contamination in feed (Commission Recommendation 2006/576/EC) and maximum limits in food (Regulation (EC) No 1881/2006), with monitoring systems to ensure food and feed safety [10].

In pigs, FB1 exposure impairs gut barrier function by disrupting tight junction proteins and inducing inflammation via pro-inflammatory cytokine production. Its hydrolyzed metabolite, HFB1, exhibits significantly lower toxicity, likely due to the removal of tricarballylic acid side chains, reducing its bioactivity [3,11]. Research demonstrates that the gut microbiota plays a functional role in this process, with specific microbes, including *Lactobacillus* spp. and certain *Bifidobacterium* spp., showing varying capacities to bind, enzymatically degrade, or metabolize mycotoxin [2,11,12,13,14]. Studies show that *Lactobacillus* spp. in the pig caecum bind FB1, primarily through physical adsorption to cell wall components, potentially supporting intestinal barrier integrity, immune modulation, and gut health [15,16]. In addition, *Lactobacillus* strains are key contributors to converting FB1 to HFB1 under diverse conditions, though other microbial genera may also play significant roles [3,17,18]. Although certain *Lactobacillus* strains adapted to the porcine gut show potential as feed additives for mitigating FB1 toxicity, their detoxification efficiency varies by strain [11]. The strain-specific mechanisms of probiotic *Lactobacillus* spp. in FB1 detoxification, such as enzymatic hydrolysis or cell wall binding, remain poorly understood [19,20,21], and studies are primarily limited to experimental conditions [22,23]. Furthermore, gut physiological conditions such as pH (5.5–7.0), temperature (~39 °C in pigs), and microbial interactions, modulate FB1 degradation by influencing enzyme activity and microbial competition [6,13,24,25]. Variations in gut ecosystem composition can significantly alter degradation kinetics, with certain bacterial strains showing a higher detoxification efficiency. These physiological conditions influence the metabolic activity, enzyme expression, and FB1 stability and bioavailability within the gastrointestinal environment.

This study investigated the microbiota-mediated hydrolysis of FB1 to its less toxic metabolite, HFB1, by *Lactobacillus* spp. isolated from the porcine caecum using an in vitro fermentation model simulating physiological conditions. By quantifying strain-specific hydrolytic efficiency and monitoring the temporal dynamics of FB1 conversion, this work provides preliminary evidence supporting the contributory role of *Lactobacillus* spp. in the detoxification of FB1 within the pig gut ecosystem. These findings highlight the metabolic potential of *Lactobacillus* species and establish a mechanistic foundation for subsequent in vivo investigations aimed at elucidating the enzymatic pathways, gene regulatory networks, and host–microbe interactions regulating *Lactobacillus*-mediated mycotoxin biotransformation in pigs.

## 2. Results

### 2.1. Bacterial Abundance and Temporal Dynamics of Gut Microbiota Incubated Without (C1: Buffer + Caecal Chyme) and with FB1 (E: Buffer + Caecal Chyme + FB1)

Analysis of bacterial abundance showed no significant differences in *Bacteroides* and *Prevotella* between C1 and E, indicating a comparable baseline microbial composition. However, *Lactobacillus* spp. exhibited greater variability in group E, while total bacterial counts varied more in C1, with a trend toward higher abundance that may reflect a treatment-induced microbial response (Figure 1).

Principal component analysis (PCA) revealed distinct clustering patterns between the C1 and E groups, indicating that FB1 exposure influenced the relative abundance and distribution of major bacterial taxa, including *Lactobacillus* spp., *Bacteroides*, and *Prevotella*. The first principal component (Dim1), accounting for 73.2% of the total variance, primarily separated the two groups, while Dim2 (24.1%) captured intra-group variability, with vectors suggesting a stronger association of *Lactobacillus* spp. and *Bacteroides* and *Prevotella* with the FB1-exposed microbiome (Figure 2).

### 2.2. Effect of Gut Microbiota on FB1

The results of the analysis of variance (ANOVA) showed that incubation time significantly influenced total bacterial abundance, as well as the abundance of *Bacteroides* and *Prevotella*, indicating dynamic ecological or metabolic changes in caecal chyme incubated without (C1: buffer + caecal chyme) or with FB1 (E: buffer + caecal chyme + FB1) (Table 1). However, neither the experimental group nor the interaction between group and time had a significant effect. In contrast, the abundance of *Lactobacillus* spp. was significantly affected by the experimental group, incubation time, and their interaction.

Results of the two-way ANOVA for the effects of group (C1: buffer + caecal chyme; E: buffer + caecal chyme + FB1), incubation time, and their interaction on bacterial abundance in caecal chyme are presented in Table 2. Both group and group × time interactions significantly influenced *Lactobacillus* spp. abundance (*p* < 0.001), suggesting group-dependent temporal responses.

The differential growth dynamics of *Bacteroides* and *Prevotella*, *Lactobacillus* spp., and total bacterial counts observed between C1 and E groups during 0, 24, and 48 h of incubation are shown in Figure 3.

### 2.3. Efficiency of Detoxification of FB1 to HFB1 (E_FB1_, %)

The efficiency of detoxification of FB1 to HFB1 (EFB1, %), which refers to the percentage of FB1 converted into its hydrolyzed metabolite HFB1 by microbial or enzymatic activity under defined conditions. Table 3 shows the concentrations of hydrolyzed fumonisin B1 (HFB1) and the efficiency of FB1 detoxification (EFB1) by gut microbiota in C2 and E groups at 0, 24, and 48 h. HFB1 was not detected in the control group (C2) at any time point, while the E group showed progressive increases, reaching 1.238 ± 0.339 µg/mL at 24 h and 1.483 ± 0.079 µg/mL at 48 h. Correspondingly, the efficiency of detoxification (EFB1) rose from 47.14% at 24 h to 56.47% at 48 h, indicating active microbial conversion of FB1 to HFB1 over time.

## 3. Discussion

The in vitro fermentation of pig caecal chyme revealed clear differences between C1 and E (Figure 1; Table 1). Total bacterial counts remained stable across incubation times in both C1 and E, suggesting that the presence of FB1 did not significantly alter overall microbial abundance but rather modulated community composition. At the genus level, *Bacteroides* and *Prevotella*, exhibited a gradual increase from 0 to 48 h, with slightly higher counts in C1 (11.6 ± 0.5 log_10_ copy number/g) compared to E (11.35 ± 0.02 log_10_ copy number/g), indicating that FB1 in E might have mildly suppressed their proliferation. Conversely, *Lactobacillus* spp. showed consistently higher counts in E across all time points, reaching 12.66 ± 0.05 log_10_ copy number/g at 48 h compared to 11.66 ± 0.48 log_10_ copy number/g in C1 (Table 1). We observed that the stabilization of the total bacterial loads after 24 h is consistent with the system’s carrying capacity. However, the nature of the response in both C1 and E is attributed to the ability of these *Lactobacillus* spp. to detoxify FB1 by converting it into another form, and their capacity to bind FB1 to their cell walls, thus preventing its absorption by other gut microorganisms [26].

The PCA revealed distinct microbial community structures between groups C1 and E, with clustering patterns indicating compositional divergence rather than differences in alpha diversity, as shown in Figure 2. The first two dimensions explained 73.2% and 24.1% of the total variance, respectively. Although total bacterial loads were unaffected by FB1 exposure, community composition was dynamically reshaped. FB1 exposure was associated with increased abundance of *Lactobacillus* spp. and a reduction in Bacteroides and *Prevotella*, suggesting selective ecological adaptation under toxin stress [3,23,27,28,29]. Several *Lactobacillus* strains have previously demonstrated antifungal activity against *Fusarium*, *Penicillium*, *Aspergillus*, and *Monilia* [26]. In the present study, seven dominant *Lactobacillus* isolates were identified, including *L. pontis* (1 strain), *L. amylovorus* (4 strains), and *L. ultunensis* (2 strains), as potential candidates involved in FB1 biodegradation. Although bacteriocin production was not directly measured, these species are known producers of antimicrobial peptides such as amylovorin *L* and helveticin *J* [30,31,32,33]. *L. pontis* also harbors genes for extracellular polysaccharide synthesis, enhancing stress tolerance and carbohydrate metabolism [32]. Given their acid and bile resistance, starch degradation [22,34,35,36], and ability to hydrolyze FB1 to HFB1 in vitro, the identified caecal *Lactobacillus* spp. in the present study show potential as host-probiotics to mitigate FB1-induced gut stress in pigs, pending in vivo validation.

The results of the analysis of variance (ANOVA) indicated that *Lactobacillus* spp. were significantly influenced by treatment, time, and their interaction (*p* < 0.001), whereas *Bacteroides* and *Prevotella* were mainly affected by incubation time (*p* < 0.001) (Table 2). Additionally, the relative abundance of *Lactobacillus* spp. increased in the FB1-exposed (E) group compared with the C1, suggesting a stress-responsive proliferation. This suggests that *Lactobacillus* spp. may contribute to the early microbial adaptation to FB1 through metabolic flexibility and potential involvement in detoxification pathways. Similar responses have been described where *Lactobacillus* species hydrolyze or adsorb mycotoxins, thereby mitigating toxicity [15,37,38]. Although *Bacteroides* and *Prevotella* dominate saccharolytic fermentation, their limited variation under FB1 exposure suggests a lower contribution to toxin transformation under in vitro conditions [15]. The microbial shifts observed reflect the adaptability of *Lactobacillus* to environmental changes and its possible role in maintaining the stability of the gut ecosystem. Since real-time pH and metabolite monitoring were not performed, this limits the mechanistic interpretation and thus, future studies integrating metabolomics are required to confirm FB1 biotransformation pathways and microbial interactions [39,40,41]. Overall, the results of this study confirm that incubation time is a key driver of compositional shifts in pig caecal microbiota under in vitro conditions during FB1 exposure, with *Lactobacillus* spp. as potential markers of treatment- or diet-induced fermentation effects [15,42]. Nonetheless, we emphasize that the observed variation in the present study is attributed to differences in bacterial community composition, substrate availability, and experimental conditions [21,43].

The hydrolysis of FB1 to its less toxic metabolite, HFB1, was observed exclusively in the FB1-treated group (E), with HFB1 concentrations increasing over the incubation period. No HFB1 was detected in the control group (C2), suggesting a microbial-mediated biotransformation process (Table 3). In the current investigation, FB1-to-HFB1 conversion efficiencies ranged from 47.14% at 24 h to 56.47% at 48 h, comparable to the 46% and 50% conversion rates reported by Fodor et al. [44], at 48 and 72 h, respectively. Discrepancies between studies may arise from methodological differences, potential overestimation of HFB1, or variations in microbial activity due to experimental conditions. Additionally, *Lactobacillus* spp. are well-known for their probiotic benefits, including improved digestion, enhanced gut barrier function, and maintenance of microbial homeostasis [3,21,28]. For instance, HFB1 conversion efficiencies reported in studies [43,44], are generally below 1%, whereas other literature indicates that *L. plantarum* and *L*. *pentosus* exhibit high detoxification efficiencies for FB1 and FB2, reaching up to 58.9% and 86.5%, respectively [21]. Additionally, *L. plantarum* has demonstrated physical adsorption-based FB1 detoxification, with efficiencies ranging from 32.9% to 61.7% within 2–4 h [15]. Although in vitro models provide a practical model for evaluating microbial toxin metabolism, extrapolating these findings to in vivo conditions should consider how host-related factors, such as intestinal motility, pH gradients, immune responses, and microbial interactions, which can modulate detoxification kinetics, will influence the animal response [44].

Following the ingestion of *Fusarium*-contaminated feed, FB1 crosses the intestinal barrier and interacts with both host cells and the gut microbiota. The bacterial cell wall components, including lipoproteins and exopolysaccharides, bind FB1 through weak intermolecular forces, thereby reducing its bioavailability. Within host tissues, FB1 inhibits ceramide synthase, disrupting sphingolipid metabolism and activating MAPK and NF-κB signaling pathways that elevate pro-inflammatory cytokines (TNF-α, IL-1β, and IL-6). Certain gut bacteria enzymatically hydrolyze FB1 through carboxylesterase or aminotransferase activity, cleaving the tricarboxylic acid side chain to form hydrolyzed FB1 (HFB1), a less toxic aminopentol derivative. These microbial detoxification processes, together with host anti-inflammatory responses (IL-10, TGF-β) and the mTORC1-mediated regulation of detoxification enzymes, contribute to restoring intestinal homeostasis by normalizing sphinganine/sphingosine ratios, strengthening tight junctions, and reducing inflammation, as presented in Figure 4.

Moreover, the mechanisms of physical and chemical adsorption are crucial for the microbial detoxification of mycotoxins in the pig gut, as reported in the existing literature [45]. Physical adsorption represents the initial and primary mode of interaction between the gut microbiota and FB1, followed by enzymatic biotransformation. In addition, previous studies have demonstrated the protective role of *Lactobacillus* species against fumonisins through two mechanisms: (1) direct mycotoxin binding and biodegradation, (2) the attenuation of mycotoxin-induced toxicity, including DNA damage and hepatotoxicity [40,46]. Collectively, this process largely relies on cell wall-mediated adsorption through weak intermolecular forces (e.g., Van der Waals forces), hydrogen bonding, or stronger chemical interactions involving surface functional groups such as hydroxyl, carboxyl, and amino groups on microbial cell walls [1,2,8,14,21,47,48]. The process begins with a rapid physical adsorption phase (often within minutes), during which mycotoxins bind to bacterial cell walls via non-covalent interactions. Subsequently, a slower phase of mass transfer and stabilization occurs, during which the bound toxins are retained on the microbial surface, facilitating later enzymatic biotransformation [21,44]. At the molecular level, two-component regulatory systems (WalKR, LiaSR, BceRS) likely sense FB1-induced membrane stress, leading to the upregulation of genes related to cell wall integrity and esterase production. Furthermore, global regulators such as CodY, Rex, PerR, OhrR, and σ^b^ may coordinate nutrient sensing, redox balance, and stress responses to support FB1 detoxification. Consequently, reduced FB1 levels could also modulate host mTORC1 signaling, mitigating stress and cytokine-driven effects on cell survival and immunity (Figure 4). However, the in vivo validation of these mechanisms remains necessary. 

Following these regulatory responses, the metabolic transformation of FB1 occurs primarily through the enzymatic cleavage of the propane-1,2,3-tricarboxylic acid side chain, converting FB1 into HFB1 [44,49] (Figure 4). This sequential process enables the efficient detoxification of FB1 in the pig gut, where adsorption and enzymatic biotransformation act synergistically to reduce mycotoxin toxicity and its harmful effects (Figure 4).

Based on current evidence, the microbial detoxification of FB1 in the pig gut can be conceptualized as a sequential, three-step process [15,48].

Adsorption: The first step involves physical interactions in which FB1 rapidly binds to microbial cell walls through non-covalent forces such as Van der Waals interactions and hydrogen bonds. This step occurs within minutes, concentrating the toxins on microbial surfaces and preventing their reabsorption into the gut lumen.Stabilization and Mass Transfer: After binding, the toxins are stabilized on microbial surfaces, reducing their dissociation back into the intestinal lumen. The bound toxins are distributed among microbial communities through diffusion and convective flow, influenced by gut dynamics over several hours to days.Enzymatic Biotransformation: In the final step, bound mycotoxins undergo enzymatic hydrolysis by microbial enzymes that cleave the propane-1,2,3-tricarboxylic acid side chain of FB1, converting it into the less toxic metabolite HFB1 [2,44,50].

Mechanistically, the detoxification sequence can be summarized as follows:i.Transport: Mycotoxins migrate through the intestinal lumen toward microbial surfaces via diffusion and convective flow, influenced by gut motility and luminal content dynamics.ii.Surface Interaction: Mycotoxins associate with microbial surface polymers, such as peptidoglycans, lipopolysaccharides, exopolysaccharides, and lipoproteins, which facilitate toxin binding to the bacterial cell walls.iii.Adsorption: Mycotoxins bind non-covalently to microbial cell walls, creating stable associations that reduce their availability for absorption in the gut.iv.Biotransformation: Microbial enzymes hydrolyze FB1 into less toxic metabolites, such as hydrolyzed HFB1 and aminopentol, thereby reducing its toxicity in the gut.

While the current study was designed in vitro to evaluate the hydrolysis of FB1 by the porcine caecal microbiome, there are several factors that limit the broader interpretation of its findings. These limitations include the use of a caecal sample from a single pig (n = 1), which, despite preliminary microbiome profiling indicating compositional consistency, prevents the assessment of inter-animal gut microbial variability and consequently restricts generalizability. In addition, the absence of external bacterial strains (e.g., commercial probiotics) or a positive control containing known FB1-degrading microorganisms (e.g., *Sphingomonas* sp.) hindered the comparative evaluation of detoxification efficiency. Furthermore, this study did not experimentally confirm the hypothesized enzymatic hydrolysis or physical adsorption mechanisms illustrated in Figure 4, thereby constraining comprehensive mechanistic interpretation. Notwithstanding, the caecal *Lactobacillus* spp. in the present study achieved a remarkable efficiency of FB1 (%E_FB1_) conversion to fully hydrolyzed FB1 (HFB1), after 24 h of incubation, suggesting a potential host-microbial contribution to detoxification. However, non-enzymatic or spontaneous degradation cannot be fully excluded. 

## 4. Materials and Methods

### 4.1. Experimental Design, Sampling and Isolating

The experimental protocol was authorized by the Food Chain Safety and Animal Health Directorate of the Somogy County Agricultural Office, under permission number XV-I-31/1509-5/2012.

Caecal content was collected from a healthy, 130 kg female Hungarian Large White pig (*Sus scrofa domesticus*), immediately post-slaughter. The animal originates from the herd maintained by the University under controlled conditions, including uniform feed, age, and housing environment, and is slaughtered in the University abattoir (University of Kaposvar, Hungary). For this exploratory in vitro study, a single animal was deemed sufficient to evaluate the conversion of FB1 to its hydrolyzed form, HFB1, and assess microbial population responses to FB1 exposure, given the minimized variability from these controlled conditions. Caecal content was aseptically collected into sterile bottles and immediately placed in anaerobic plastic bags with Anaerocult gas generators (Merck, Darmstadt, Germany) to maintain an anaerobic environment during transport to the laboratory.

To isolate *Lactobacillus* spp., 1 g of caecal content was collected using a sterile Pasteur pipette and mixed with 9 mL of MRS (de Man, Rogosa, and Sharpe) broth purchased from (BioLab, Budapest, Hungary) to create a 1:10 dilution. This mixture was then plated (100 µL) onto the surface of MRS agar plates (BioLab). The plates were incubated at 37 °C for 3 days under anaerobic conditions to allow *Lactobacillus* colonies to grow. The plating procedure was repeated in duplicate. After 3 days, the colonies were identified and transferred to fresh MRS agar plates to ensure purity and incubated for another 3 days under the same conditions. Once isolated, the *Lactobacillus* strains were divided into two parts: One portion was stored at −80 °C until sequencing, and the other portion was suspended in 50 mL of fresh MRS broth and incubated at 37 °C for 3 days to allow for bacterial growth. After incubation, 2 mL of this culture was stored at −80 °C for bacterial concentration analysis using qPCR. Subsequently, McDougall buffer solution was prepared by dissolving 9.8 g of NaHCO_3_, 3.7 g of anhydrous Na_2_HPO_4_, 0.57 g of KCl, 0.47 g of NaCl, 0.12 g of MgSO_4_·7H_2_O, and 0.04 g of CaCl_2_ in 1000 mL of distilled water. The solution was adjusted to pH 8.3 and pre-incubated at 37 °C. This buffer was used both as a control solution and as a medium to homogenize the caecal chyme samples.

Three treatment groups were established: (i) Control 1 (C1: buffer + caecal chyme), (ii) Control 2 (C2: buffer + FB1), and (iii) Experimental group (E: buffer + caecal chyme + FB1) (Table 4). The C2 was set up to evaluate the efficiency of FB1 conversion to HFB1 in comparison to E. Each group consisted of 12 samples, with four samples collected at each time point (0, 24, and 48 h). A 3.33 g aliquot of caecal chyme was suspended in pre-incubated McDougall buffer tubes designated for the C1 and E groups. After a 4 h pre-incubation at 37 °C, FB1 purchased from Sigma-Aldrich (Darmstadt, Germany) was added to the tubes in the C2 and E groups to achieve a final concentration of 5 µg/mL (Table 4).

At each time point (0, 24, and 48 h), 4 samples were collected from each group, for a total of 12 samples per treatment group. Samples were incubated anaerobically and then centrifuged at 2000× *g* for 20 min. The supernatants were used for bacterial concentration analysis by qPCR, as well as for measuring FB1 and its hydrolyzed form (HFB1) concentrations using Liquid chromatography-mass spectrometry (LC-MS).

### 4.2. DNA Sequencing: Extraction, Quantification, and qPCR

The DNA extraction was performed on 200 µL of frozen MRS broth containing *Lactobacillus* strains using the QIAamp^®^ DNA Stool Mini Kit (Roche Diagnostics GmbH, Mannheim, Germany) following the manufacturer’s instructions. Total *lactobacilli* and *coliform* were detected by real-time qPCR. The PCR condition was identical for Total bacteria, *Bacteroides* and *Prevotella*, and *Lactobacillus* spp., it consisted of an initial 10 min. at 95 °C, followed by 40 cycles of 30 s at 95, 1 min at 60 °C. The primer sequences used for the PCR are listed in Table 5.

All samples were analyzed in triplicate. The bacterial content of each sample was quantified using qPCR (SYBR Green), based on a standard curve derived from a dilution series of purified PCR products for *Lactobacillus* spp., whereas the dilution series of a plasmid containing the PCR product was used to prepare the standard curve for total bacteria, and *Bacteroides* and *Prevotella.*

In the case of *Lactobacillus* spp., a control sequence for the standard curve was created using the primers TGTAAAACGACGGCCAGT-AGCAGTAGGGAATCTTCCA (forward) and CAGGAAACAGCTATGACC-CACCGCTACACATGGAG (reverse), with M13 sequences were incorporated into the *Lactobacillus*-specific primers. The M13 sequence was used to confirm the control fragment sequence through sequencing. We validated the accuracy and efficiency of the PCR assays using the template DNA of the *Lactobacillus* reference species as described in [55].

Sequencing was carried out using a BigDye^®^ Direct Cycle Sequencing Kit, USB^®^ ExoSAP-IT^®^ PCR Product Cleanup, and BigDye^®^ XTerminator™ Purification Kit (Applied Biosystems, Foster City, CA, USA), followed by analysis with the Applied Biosystems 3500 Genetic Analyzer. The nucleotide sequences were identified using the Nucleotide BLAST algorithm (https://blast.ncbi.nlm.nih.gov/Blast.cgi (accessed on 15 November 2018) and the 16S ribosomal RNA sequence database (Bacteria and Archaea).

### 4.3. Extraction of FB1, HFB1 and LC-MS Analysis Protocol

For FB1 extraction, samples from the experimental and control-2 groups were diluted 1:1 (7 mL sample + 7 mL distilled water) and centrifuged at 3000 rpm for 5 min. The supernatant was processed using a modified Sep-Pak C18 cartridge protocol (Waters Co., Milford, MA, USA) as described by [20,44]. Briefly, columns were preconditioned with 2 mL of methanol followed by 2 mL of distilled water. A 2 mL aliquot of the diluted sample was loaded onto the columns, which were then washed with 2 mL of distilled water. FB1 was eluted with 2 mL of a 1:1 (*v*/*v*) water/acetonitrile mixture.

Liquid chromatography–mass spectrometry (LC-MS) analysis was performed using a Shimadzu Prominence UFLC system coupled to an LC-MS 2020 single quadrupole mass spectrometer (Shimadzu, Kyoto, Japan) with an electrospray ionization source. Mass spectra were optimized with an interface voltage of 4.5 kV, a detector voltage of 1.05 kV in negative mode, and 1.25 kV in positive mode. Separation was achieved using a Phenomenex Kinetex 2.5 µm C18 (2)-HST column (100 × 2.0 mm, Lane Cove, NSW, Australia) at 40 °C with a flow rate of 0.3 mL/min. Gradient elution employed LC-MS grade water (VWR Hungary, Debrecen) as eluent A and acetonitrile eluent B, both acidified with 0.1% acetic acid. Subsequently, a 10 µL sample was injected, and the following gradient was applied: 0 min, 5% B; 3 min, 60% B; 8 min, 100% B; held at 100% B for 3 min; and re-equilibrated at 5% B for 2.5 min. FB1 and HFB1 standards (diluted from 1000 mg/L and 25 mg/L, respectively) were used as references. MS parameters were source block temperature 90 °C, dissolving temperature of 250 °C, heat block temperature of 200 °C, and drying gas flow of 15 L/min. Detection was performed in selected ion monitoring (SIM) mode.

### 4.4. Statistical Analysis

Data were analyzed using R version 4.4.1 [56]. To evaluate the effects of Groups and Time_hours on the abundance of each bacterial group (Total bacteria, *Bacteroides* and *Prevotella*, and *Lactobacillus* spp.), a two-way analysis of variance (ANOVA) was performed using the model: *anova_groups<*− *aov(log_10_(groups + 1) ~ Groups * Time_hours,* to compare C1 and E groups. The model included Groups, Time_hours, and their interaction as fixed effects.

A post hoc test was conducted to perform pairwise comparisons between groups. The Tukey HSD test was applied using a 95% family-wise confidence level to control for multiple comparisons and perform only if the interaction term is significant using: *if (summary(anova_groups)[[1]]$`Pr(>F)`[3] < 0.05)*. Boxplots were generated using the ggplot2 package. Bacterial count data were log-transformed using $log_{10}(Count+1)$ to normalize the data. The data were grouped by experimental conditions (Groups) and bacteria type (Bacteria_Type). Data log-transformed for normalization, and facet_wrap() were employed to separate each bacterial species for clarity.

To evaluate shifts in bacterial community structure between the C1 and E groups over time, Principal Component Analysis (PCA) was performed on log-transformed bacterial abundance data $\left(log_{10}(Count+0.001)\right)$ using the syntax of *prcomp()* function, total variance values were obtained using the *summary()* function applied to the *prcomp()* output. The PCA plot was generated using the *fviz_pca_ind()* function from the factoextra package, with the addEllipses = TRUE option to display confidence ellipses around E groups based on time (0, 24, and 48 h). The analysis aimed to capture microbial compositional differences in response to FB1 exposure and how bacterial composition shifts across groups (C1 vs. E) and over time (0 h, 24 h, 48 h)**.**

The percentage efficiency (%E_FB1_) of FB1 conversion to fully hydrolyzed FB1 (HFB1) was calculated for the C2 and E groups based on the molecular weights of FB1 (721 g/mol) and HFB1 (405 g/mol) using the following formula:$$\%E_{\mathrm{FB1}}\;=\;\frac{Hydrolyzed\;fumonisin\;B1\;(mol/g)\times 721\;g/mol}{405\;g/mol\times fumonisin\;B1\;(mol/g)}\times \;100$$

## 5. Conclusions

Porcine caecal Lactobacillus spp. exhibited a notable efficiency in converting FB1 (%EFB1) to fully hydrolyzed FB1 (HFB1) after 24 h of incubation, indicating a possible host-microbial role in detoxification in vitro. However, non-enzymatic or spontaneous degradation cannot be entirely ruled out. Additionally, the observed increase in *Lactobacillus* abundance in this study suggests resilience to FB1, but the use of only single animals limits the ability to generalize mechanistic insights into hydrolysis and adsorption in vivo. Consequently, further in vivo research is needed to establish causality, account for individual animal variation, and determine the ecological significance of FB1 detoxification processes under practical feeding conditions in pigs.

## Figures and Tables

**Figure 1 ijms-26-10557-f001:**
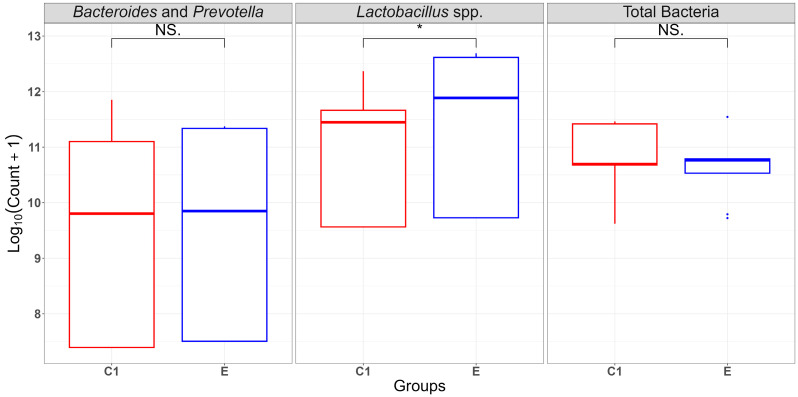
Effects of FB1 on the abundance of *Bacteroides* and *Prevotella*, *Lactobacillus* spp., and total bacteria count $\left(log_{10}\left(Count+1\right)\right)$ in C1 (buffer + caecal chyme) and E (buffer + caecal chyme + FB1). * Indicates statistically significant differences (*p* < 0.001); NS denotes non-significant effects.

**Figure 2 ijms-26-10557-f002:**
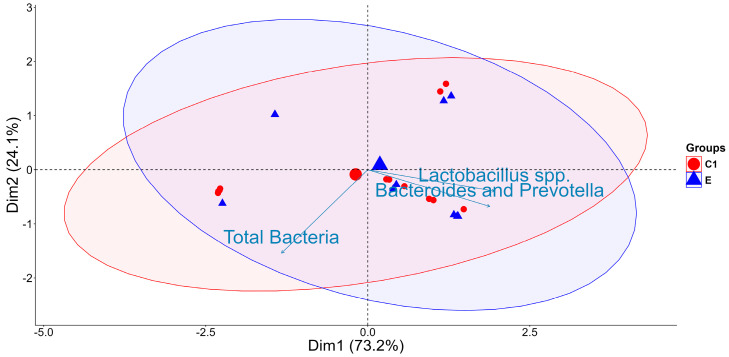
Principal component analysis (PCA) of qPCR-derived abundances of *Bacteroides* and *Prevotella*, *Lactobacillus* spp., and total bacteria in caecal chyme incubated without (C1: buffer + caecal chyme) or with FB1 (E: buffer + caecal chyme + FB1). Red and blue ellipses indicate the confidence intervals around the centroids of C1 and E, respectively.

**Figure 3 ijms-26-10557-f003:**
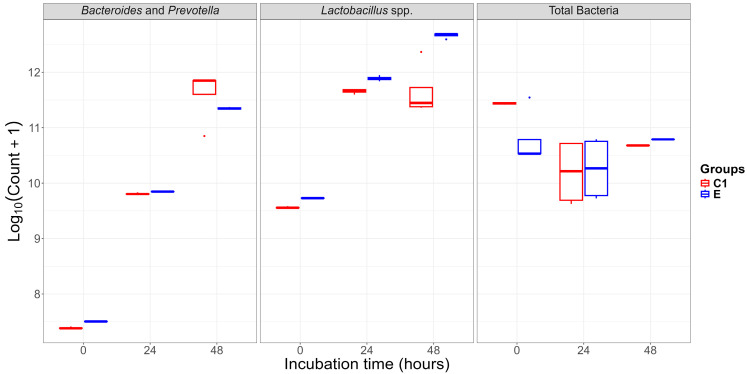
Box plots showing the log-transformed bacterial counts $\left(log_{10}(Count+1)\right)$ at 0, 24, and 48 h of different bacterial groups: *Bacteroides* and *Prevotella*, *Lactobacillus* spp., and Total Bacteria.

**Figure 4 ijms-26-10557-f004:**
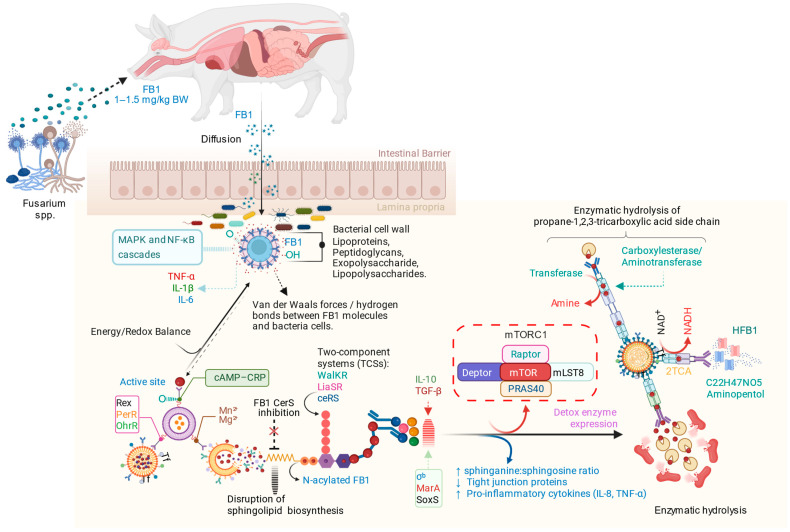
Overview of the assumed mechanism of fumonisin B1 (FB1) biodegradation by gut microbes.

**Table 1 ijms-26-10557-t001:** Bacterial abundance in caecal chyme incubated without (C1: buffer + caecal chyme) or with FB1 (E: buffer + caecal chyme + FB1), at 0, 24, and 48 h, measured using qPCR ($log_{10}copynumber/g,mean\pm SD$).

	0 h		24 h		48 h	
	Log10 Copy Number/g					
Bacteria Groups	C1	E	C1	E	C1	E
Total Bacteria	11.44 ± 0.02 ^aA^	10.78 ± 0.51 ^aB^	10.19 ± 0.61 ^bA^	10.26 ± 0.58 ^bA^	10.68 ± 0.00 ^bA^	10.79 ± 0.00 ^bA^
*Bacteroides* and *Prevotella*	7.39 ± 0.01 ^Ab^	7.5 ± 0.00 ^aA^	9.8 ± 0.01 ^bA^	9.85 ± 0.00 ^bA^	11.6 ± 0.50 ^cA^	11.35 ± 0.02 ^cA^
*Lactobacillus* spp.	9.56 ± 0.01 ^aB^	9.73 ± 0.00 ^aA^	11.66 ± 0.04 ^bA^	11.89 ± 0.05 ^bA^	11.66 ± 0.48 ^bB^	12.66 ± 0.05 ^cA^

^A,B^ Different uppercase superscripts denote a significant (*p* < 0.01) difference between C1 and E groups at the same point. ^a,b,c^ denote significant (*p* < 0.01) differences between incubation times within groups.

**Table 2 ijms-26-10557-t002:** ANOVA Results for Total Bacteria, *Bacteroides* and *Prevotella*, and *Lactobacillus* spp. caecal chyme incubated without (C1: buffer + caecal chyme) or with FB1 (E: buffer + caecal chyme + FB1).

Total *Bacteria*					
Source of Variation	df	Sum of Squares	Mean Square	*F*-Value	*p*-Value
Groups	1	0.15	0.15	0.934	0.35
Time (h)	2	3.162	1.581	9.849	**0.001**
Groups: Time (h)	2	0.741	0.37	2.307	0.13
Residuals	18	2.89	0.161		
***Bacteroides* and *Prevotella***					
Groups	1	0	0	0.119	0.73
Time_hours	2	65.73	32.86	786.751	**<0.001**
Groups: Time (h)	2	0.15	0.08	1.816	0.19
Residuals	18	0.75	0.04		
***Lactobacillus* spp.**					
Groups	1	1.327	1.327	33.79	**<0.001**
Time (h)	2	29.418	14.709	374.46	**<0.001**
Groups: Time (h)	2	0.871	0.435	11.08	**<0.001**
Residuals	18	0.707	0.039		

In case of significant differences (*p* < 0.05), *p* values are bold.

**Table 3 ijms-26-10557-t003:** Measurement of Hydrolyzed Fumonisin B1, HFB1 concentration and efficiency of gut microbiota in converting FB1 to HFB1 in control 2 (C2: buffer + FB1), and experimental (E: buffer + caecal chyme + FB1) groups at 0, 24 and 48 h.

Item	Sampling Time, Hours	HFB1 Concentrations, µg/mL		E_FB1_, %
		C2	E	
	0	0	0	
HFB1	24	0	1.238 ± 0.339	47.14
	48	0	1.483 ± 0.079	56.47

E_FB1_ = Efficiency of conversion of FB1 to HFB1 (%).

**Table 4 ijms-26-10557-t004:** Experimental design to determine in vitro interaction between fumonisin B1 and the intestinal microflora of pigs.

Item	Treatments		
	C1	C2	E
Buffer, mL	5.67	5.67	5.67
Caecal chyme, g	3.33	-	3.33
Distilled H_2_O, mL	1	-	-
FB1 Concentrations, µg/mL	-	5	5
Samples per treatment and incubation time	4	4	4
Incubation time, hours	0, 24, 48	0, 24, 48	0, 24, 48

Each group contained 12 samples, with 4 samples collected at each of the three time points (0, 24, and 48 h).

**Table 5 ijms-26-10557-t005:** Primer Sequences.

Bacteria	Primer Sequences		Amplicon Size (bp)	References
	Forward (5′-3′)	Reverse (5′-3′)		
Total Bacteria	GCAGGCCTAACACATGCAAGTC	CTGCTGCCTCCCGTAGGAGT	292	[51,52]
*Bacteroides* and *Prevotella*	GAAGGTCCCCCACATTG	CAATCGGAGTTCTTCGTG	418	[53]
*Lactobacillus* spp.	AGCAGTAGGGAATCTTCCA	CACCGCTACACATGGAG	340	[54]

## Data Availability

The original contributions presented in this study are included in the article. Further inquiries can be directed to the corresponding author.

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
