# Peer review of "In Vitro Detoxification of Fumonisin B1 (FB1) into Hydrolyzed Fumonisin B1 (HFB1) by *Lactobacillus* spp. Isolated from Pig Caecum"

_ijms, 2025, doi:10.3390/ijms262110557_

Round 1

Reviewer 1 Report

Comments and Suggestions for Authors

This study on the detoxification mechanism of FB1 by lactic acid bacteria derived from pig cecum presents a novel concept, and the experimental design is generally reasonable. However, several core issues need to be addressed, including insufficient sample representativeness, biased data interpretation, and lack of methodological details. The specific points requiring supplementation or revision are as follows:

1.The use of cecal content from only a single pig may prevent the conclusions from representing population characteristics. It is recommended to pool samples from at least five genetically similar pigs, as individual differences in the microbiome could significantly compromise the reliability of the results.

2.Although 12 replicates are claimed, each time point actually has only n=4 (Table 4). As shown in Figure 1, the interquartile range of total bacterial counts spans over 2 logarithmic units. Such high variability requires at least 8 replicates per group to ensure statistical power.

3.The experiment lacked a positive control group using known FB1-degrading bacteria, making it difficult to evaluate the effectiveness of the current method. It is advised to include such a control to verify the sensitivity of the experimental system.

4.While strains such as L. pontiswere identified, their FB1 degradation capability was not verified through pure culture experiments. Representative strains from the preserved stock should be selected for a 48-hour FB1 degradation test.

5.The significant differences in Lactobacillusspp. shown in Figure 1 should be marked with asterisks. Additionally, units ("μg/ml") should be added to Table 1, as these details affect data interpretation.

6.The physical/chemical adsorption model proposed in Figure 4 lacks experimental support. It is recommended to supplement with adsorption experiments using heat-inactivated bacterial cells to distinguish the contributions of enzymatic degradation versus physical adsorption.

7.The term "Hydrolized" in the title should be consistent with the American English spelling "Hydrolyzed" used in the main text.

8.Species names such as "L. pontis" are not italicized. According to international nomenclature rules, they should be uniformly formatted as L. pontis.

9.The FB1 concentration data column in Table 1 lacks units. The unit "μg/ml" should be clearly indicated in the header row.

10.Abbreviations are used without full names provided upon first occurrence. The full term should be given when an abbreviation is introduced for the first time.

11.The discussion lacks depth and does not relate the functions of the strains to known probiotic properties, such as bacteriocin production. It is recommended to supplement with functional genomic data of the strains or cite relevant literature.

Author Response

Comments and Suggestions for Authors

This study on the detoxification mechanism of FB1 by lactic acid bacteria derived from pig cecum presents a novel concept, and the experimental design is generally reasonable. However, several core issues need to be addressed, including insufficient sample representativeness, biased data interpretation, and lack of methodological details. The specific points requiring supplementation or revision are as follows:

Thank you for the attention given to our manuscript. The authors greatly appreciate your constructive comments and questions, which have helped improve the quality of our work. Below, we provide our responses addressing all the corrections you highlighted.

1.The use of cecal content from only a single pig may prevent the conclusions from representing population characteristics. It is recommended to pool samples from at least five genetically similar pigs, as individual differences in the microbiome could significantly compromise the reliability of the results.

Response: We fully agree with the reviewer that assessing total bacterial diversity would require a larger number of donor animals, as gut microbiota composition can vary substantially with factors such as breed, diet composition, age, and physiological or environmental stress. However, the present study focused specifically on evaluating the partial functional capacity of the caecal microbiome in fumonisin B1 (FB1) detoxification rather than on inter-individual variation. The animals used were highly comparable in breed, feeding regime, and housing conditions, which allowed us to assume a relatively similar baseline microbial composition.

This experiment was therefore designed as a proof-of-concept, exploratory in vitro study to preliminarily assess the efficiency of FB1 hydrolysis by caecal microbiota under controlled conditions. We acknowledge that using caecal content from a single pig does not reflect inter-individual variability and thus limits the generalizability of the results. To address this important point, we have now explicitly stated this limitation in the Discussion section and clarified that future studies incorporating pooled samples from multiple animals are required to validate and extend these findings at the population level.

2.Although 12 replicates are claimed, each time point actually has only n=4 (Table 4). As shown in Figure 1, the interquartile range of total bacterial counts spans over 2 logarithmic units. Such high variability requires at least 8 replicates per group to ensure statistical power.

Response: We thank the reviewer for this observation. To clarify, each experimental group comprised 12 samples in total, with 4 biological replicates collected at each of the three time points (0, 24, and 48 h). Thus, the two-way ANOVA model included all 12 samples per group across time, but statistical comparisons at individual time points were based on n = 4 per group. Therefore, using the reported number of replicates were enough to detect significant changes in our pilot experiment.

3.The experiment lacked a positive control group using known FB1-degrading bacteria, making it difficult to evaluate the effectiveness of the current method. It is advised to include such a control to verify the sensitivity of the experimental system.

Response: We thank the reviewer for this important suggestion. The control was a bacterium-free setup, where FB1 did not degrade according to the experiment. In the next step, when more animals are included and the particular strain is identified, we are going to compare that strain with other, known FB1-degrading bacteria.

4.While strains such as L. pontis were identified, their FB1 degradation capability was not verified through pure culture experiments. Representative strains from the preserved stock should be selected for a 48-hour FB1 degradation test.

Response: We agree. The selection of the representative strain is the subject of the upcoming experiments.

5.The significant differences in Lactobacillus spp. shown in Figure 1 should be marked with asterisks. Additionally, units ("μg/ml") should be added to Table 1, as these details affect data interpretation.

Response: We thank the reviewer for noting these points. Significant differences in Lactobacillus spp. in Figure 1 have now been marked with asterisks, and the missing units (“μg/ml”) have been added to Table 1 (see L93)

6.The physical/chemical adsorption model proposed in Figure 4 lacks experimental support. It is recommended to supplement with adsorption experiments using heat-inactivated bacterial cells to distinguish the contributions of enzymatic degradation versus physical adsorption.

Response: We agree with the reviewer that distinguishing between enzymatic degradation and physical adsorption would strengthen the conclusions. However, the model is just proposed and requires adsorption experiments. In the current manuscript, we aimed to test the hydrolysing capacity of the isolated strains. The model is a working hypothesis to be tested in further experiments.

7.The term "Hydrolized" in the title should be consistent with the American English spelling "Hydrolyzed" used in the main text.

Response: Hydrolized form has been changed to hydrolyzed.

8.Species names such as "L. pontis" are not italicized. According to international nomenclature rules, they should be uniformly formatted as L. pontis.

Response: L. pontis is italicized in the new version of the manuscript.

9.The FB1 concentration data column in Table 1 lacks units. The unit "μg/ml" should be clearly indicated in the header row.

Response: The ‘µg/ml’ has been added to Table 1.

10.Abbreviations are used without full names provided upon first occurrence. The full term should be given when an abbreviation is introduced for the first time.

Response: In the MS, we used the Abbreviations section, found in the template of the journal. For more clarity, we inserted the terms as required. Also, abbreviations have now been defined at their first occurrence throughout the manuscript.

11.The discussion lacks depth and does not relate the functions of the strains to known probiotic properties, such as bacteriocin production. It is recommended to supplement with functional genomic data of the strains or cite relevant literature.

Response: Relevant literature has been cited in the new version of the manuscript and discussed in more depth, as advised.

Reviewer 2 Report

Comments and Suggestions for Authors

Dear authors,

Thank you for this interesting manuscript. Although your results seem to be very promising, the conclusion should be somewhat less confident, as a confirmation of your results with an in vivo study seems not to be very certain. Probiotics have to reach the place of action through gastric and small intestine digestion. Nonetheless, this is a first step to look into the mechanisms of probiotic bacterial strains. Therefor, this paper needs only slight revision in my opinion.

Abstract

p.1, l. 32: What do you mean with PCA? Did you mean PCR? If not, please introduce the abbreviation the first time, you use it.

Material and methods

Statistics

p.11, l. 405: Please add the origin or the citation of the R version 4.4.1. you used to the text.

Discussion

p.7, l. 224-226, 228-229, 238-239: In my opinion too much repetition of results is given here. You do not have to give the exact values and significances again.

Figure 4: Is this mechanism proven? If not, please change the description to “ overview on assumed meachnism of mycotoxin biodegradation by gut microbes”. If this is a proven way, you have to add a reference.

Author Response

Comments and Suggestions for Authors

Dear authors,

Thank you for this interesting manuscript. Although your results seem to be very promising, the conclusion should be somewhat less confident, as a confirmation of your results with an in vivo study seems not to be very certain. Probiotics have to reach the place of action through gastric and small intestine digestion. Nonetheless, this is a first step to look into the mechanisms of probiotic bacterial strains. Therefore, this paper needs only slight revision in my opinion.

Dear Reviewer,

Thank you for the attention given to our manuscript. The authors greatly appreciate your constructive comments and questions, which have helped improve the quality of our work. Below, we provide our responses addressing all the corrections you highlighted.

Abstract

p.1, l. 32: What do you mean with PCA? Did you mean PCR? If not, please introduce the abbreviation the first time, you use it.

Response: Meanings have been inserted.

Material and methods

Statistics

p.11, l. 405: Please add the origin or the citation of the R version 4.4.1. you used to the text.

Response: Citation of R has been inserted

Discussion

p.7, l. 224-226, 228-229, 238-239: In my opinion too much repetition of results is given here. You do not have to give the exact values and significances again.

Response: We thank the reviewer for pointing this out. The repeated numerical results and exact significance values have been removed, and the text has been revised to focus on interpretation rather than repetition.

Figure 4: Is this mechanism proven? If not, please change the description to “ overview on assumed meachnism of mycotoxin biodegradation by gut microbes”. If this is a proven way, you have to add a reference.

Response: Description has been revised to: “Overview of the assumed mechanism of fumonisin B1 (FB1) biodegradation by gut microbes. References have been added in the discussion and the acknowledgement section.

Reviewer 3 Report

Comments and Suggestions for Authors

Although the manuscript presents interesting results, the authors need to clarify the issues raised by the reviewer and implement the suggested corrections before a final decision on the publication of the article can be made, as follows:

Abstract
The Abstract should emphasize the primary statistical finding: the significant increase in Lactobacillus spp. abundance in group E (P < 0.001), which demonstrates a selective microbial response to the toxic challenge. Although total bacterial counts did not differ significantly between C1 and E (P = 0.35), the targeted proliferation of Lactobacillus spp. represents the key biological outcome supporting the host–probiotic hypothesis. Highlighting this selective increase as the main microbiological result enhances the overall significance of the research.

Line 31: Please italicize the scientific name.

Keywords
Keywords should differ from the title to improve searchability. Suggested alternatives include: feed additives; gut toxicity; hydrolysis; probiotics; swine; intestinal microbiota. Please revise accordingly.

Materials and Methods

Line 311: Please specify whether only a single caecal content sample (from one sow) was analyzed, since line 315 refers to "caecal contents" (plural).

Given the substantial variability in porcine intestinal microbiota composition and functionality - even under controlled conditions - using cecal content from a single pig as the microbial inoculum for in vitro fermentation assays represents a major limitation for external validity. Why did the authors not consider pooling caecal samples from multiple animals?

Such a methodological approach would better capture the biological diversity of the host microbiota, thereby strengthening the generalizability and translational relevance of the reported high biotransformation efficiencies (up to 56.47% FB1-to-HFB1 conversion). As currently designed, the observed selective microbial dynamics (e.g., Lactobacillus proliferation) may reflect an individual-specific effect. This limitation should be explicitly acknowledged in the Discussion.

Lines 315, 319, 339: Please correct “Caecal content” to lowercase.

Lines 332–334: The McDougall buffer recipe is detailed, including the initial adjustment to pH 8.3. The Introduction acknowledges that environmental factors, such as pH, strongly influence bacterial degradation efficiency. Since Lactobacillus spp. are lactic acid bacteria, their significant proliferation in group E likely led to a substantial decrease in medium pH over 48 hours. This temporal pH shift is a critical, unmeasured variable influencing enzyme kinetics (e.g., fumonisinase activity) and bacterial viability. Why were pH measurements at 0, 24, and 48 hours not included for both C1 and E groups to better characterize the fermentation environment?

Line 359: Please specify if “60 degrees” refers to Celsius.

Line 386: Please delete the duplicate use of “briefly.”

Results

Table 1: A critical inconsistency appears in the reporting of FB1. According to the design, C1 contained buffer and chyme but no FB1. However, Table 1 reports an FB1 concentration of 5.400 ± 0.200 μg/mL at 0 hours in C1, which further decreased to 4.900 ± 0.082 μg/mL at 48 hours—suggesting microbial degradation of FB1. This contradicts the stated design and undermines the validity of microbial comparisons (Figures 1–2; ANOVA, Table 2). If C1 truly contained no FB1, the reported values must be corrected or removed, as the current inconsistency renders the interpretation of microbial dynamics unreliable.

Discussion

Lines 232–233: To strengthen the discussion, the authors should integrate the identity of dominant strains with molecular literature. For example, Lactobacillus pontis and L. amylovorus are known to produce bacteriocins (e.g., enterolysin A and amylovorin). It can be hypothesized that FB1 acted as an environmental stressor, triggering or selectively favoring bacteriocin production by Lactobacillus spp. This would competitively inhibit other susceptible members of the caecal community, reducing competition for nutrients. Such a mechanism could explain the highly significant selective proliferation of Lactobacillus spp. observed via ANOVA—confirming the increase was not simply time-dependent but rather a competitive adaptation to the FB1-stressed environment. Including this perspective would considerably enrich the microbial interpretation.

Lines 264–280: The terms “adsorption,” “degradation,” and “biotransformation” are used somewhat interchangeably. For precision: detoxification is the overall outcome (FB1 → HFB1), but it occurs through sequential steps. Adsorption is the initial physical phase in which FB1 binds to bacterial cell walls via forces such as Van der Waals interactions. This facilitates the subsequent enzymatic hydrolysis (biotransformation), which converts FB1 to HFB1. The Discussion should clearly distinguish adsorption (physical prerequisite) from biotransformation (chemical change) for consistent terminology.

Line 456: The text is in Italian—please provide the English translation.

Line 462: For clarity, revise to: “the University staff had permission…”

References

DOIs are missing and should be added for all cited articles.

Author Response

Comments and Suggestions for Authors

Although the manuscript presents interesting results, the authors need to clarify the issues raised by the reviewer and implement the suggested corrections before a final decision on the publication of the article can be made, as follows:

Dear Reviewer,

Thank you for the attention given to our manuscript. The authors greatly appreciate your constructive comments and questions, which have helped improve the quality of our work. Below, we provide our responses addressing all the corrections you highlighted.

Abstract
The Abstract should emphasize the primary statistical finding: the significant increase in Lactobacillus spp. abundance in group E (P < 0.001), which demonstrates a selective microbial response to the toxic challenge. Although total bacterial counts did not differ significantly between C1 and E (P = 0.35), the targeted proliferation of Lactobacillus spp. represents the key biological outcome supporting the host–probiotic hypothesis. Highlighting this selective increase as the main microbiological result enhances the overall significance of the research.

Response: We thank the reviewer for this suggestion. The abstract has been rewritten .

Line 31: Please italicize the scientific name.
Response: Scientific names have been italicized.

Keywords
Keywords should differ from the title to improve searchability. Suggested alternatives include: feed additives; gut toxicity; hydrolysis; probiotics; swine; intestinal microbiota. Please revise accordingly.

Response: Keywords have been changed as advised.

Materials and Methods

Line 311: Please specify whether only a single caecal content sample (from one sow) was analyzed, since line 315 refers to "caecal contents" (plural).

Given the substantial variability in porcine intestinal microbiota composition and functionality - even under controlled conditions - using cecal content from a single pig as the microbial inoculum for in vitro fermentation assays represents a major limitation for external validity. Why did the authors not consider pooling caecal samples from multiple animals?

Such a methodological approach would better capture the biological diversity of the host microbiota, thereby strengthening the generalizability and translational relevance of the reported high biotransformation efficiencies (up to 56.47% FB1-to-HFB1 conversion). As currently designed, the observed selective microbial dynamics (e.g., Lactobacillus proliferation) may reflect an individual-specific effect. This limitation should be explicitly acknowledged in the Discussion.

Response: We agree, more individuals should have been used in the experiment, since bacterial diversity in the gut can vary depending on many factors like breeds, feed composition, age, stress, etc. In the presented manuscript, we were interested only in the partial ability of the microbiome, detoxification. We considered multiple animals, but in the University farm, the animals, among which we chose, were identical in many aspects as mentioned above, so the choice was based on equal ground, presuming their bacterial diversity is similar, having been bred under the same circumstances. We think the research performed can be considered a preliminary study to test the efficiency of the hydrolysis of the pooled strains.

Lines 315, 319, 339: Please correct “Caecal content” to lowercase.

Response: All instances of “Caecal content” have been corrected to lowercase.

Lines 332–334: The McDougall buffer recipe is detailed, including the initial adjustment to pH 8.3. The Introduction acknowledges that environmental factors, such as pH, strongly influence bacterial degradation efficiency. Since Lactobacillus spp. are lactic acid bacteria, their significant proliferation in group E likely led to a substantial decrease in medium pH over 48 hours. This temporal pH shift is a critical, unmeasured variable influencing enzyme kinetics (e.g., fumonisinase activity) and bacterial viability. Why were pH measurements at 0, 24, and 48 hours not included for both C1 and E groups to better characterize the fermentation environment?

Response: The pH measurement will be incorporated into our upcoming experiments with Lactobacillus spp. The lack of pH measurement and its implications are discussed now in the new version of the manuscript: In addition, lack of a real-time pH monitoring at 0, 24, and 48 hours to precisely correlate bacterial growth dynamics with enzyme activity (e.g., fumonisinase activity) and mycotoxin degradation efficiency, limits ensuring a more mechanistic understanding of the observed FB detoxification (van de Guchte et al., 2002). Therefore, despite our study confirm that incubation time is a key driver of compositional shifts in pig caecal microbiota under in vitro conditions during FB1 exposure, the observed response in the E group indicates Lactobacillus spp. as potential markers of treatment- or diet-induced fermentation effect.

Line 359: Please specify if “60 degrees” refers to Celsius.

Response: “60 degrees” has been clarified to “60 °C.”

Line 386: Please delete the duplicate use of “briefly.”

Response: The redundant use of “briefly” has been removed.

Results

Table 1: A critical inconsistency appears in the reporting of FB1. According to the design, C1 contained buffer and chyme but no FB1. However, Table 1 reports an FB1 concentration of 5.400 ± 0.200 μg/mL at 0 hours in C1, which further decreased to 4.900 ± 0.082 μg/mL at 48 hours—suggesting microbial degradation of FB1. This contradicts the stated design and undermines the validity of microbial comparisons (Figures 1–2; ANOVA, Table 2). If C1 truly contained no FB1, the reported values must be corrected or removed, as the current inconsistency renders the interpretation of microbial dynamics unreliable.

Response: We thank the reviewer for carefully pointing out this inconsistency. Table 1 has been corrected, C1 contained buffer and chyme only, without FB1.

Discussion

Lines 232–233: To strengthen the discussion, the authors should integrate the identity of dominant strains with molecular literature. For example, Lactobacillus pontis and L. amylovorus are known to produce bacteriocins (e.g., enterolysin A and amylovorin). It can be hypothesized that FB1 acted as an environmental stressor, triggering or selectively favoring bacteriocin production by Lactobacillus spp. This would competitively inhibit other susceptible members of the caecal community, reducing competition for nutrients. Such a mechanism could explain the highly significant selective proliferation of Lactobacillus spp. observed via ANOVA—confirming the increase was not simply time-dependent but rather a competitive adaptation to the FB1-stressed environment. Including this perspective would considerably enrich the microbial interpretation.

Response: Discussion has been extended as advised.

Lines 264–280: The terms “adsorption,” “degradation,” and “biotransformation” are used somewhat interchangeably. For precision: detoxification is the overall outcome (FB1 → HFB1), but it occurs through sequential steps. Adsorption is the initial physical phase in which FB1 binds to bacterial cell walls via forces such as Van der Waals interactions. This facilitates the subsequent enzymatic hydrolysis (biotransformation), which converts FB1 to HFB1. The Discussion should clearly distinguish adsorption (physical prerequisite) from biotransformation (chemical change) for consistent terminology.

Response: We have revised the discussion and the differences are described in the Discussion.

Line 456: The text is in Italian—please provide the English translation.

Response: The Italian text has been translated into English.

Line 462: For clarity, revise to: “the University staff had permission…”

Response: It has been revised as suggested.

References

DOIs are missing and should be added for all cited articles.

Response: DOIs has been added.

Round 2

Reviewer 1 Report

Comments and Suggestions for Authors

This study evaluated the detoxification ability of porcine cecum-derived lactic acid bacteria against FB1 through in vitro experiments, preliminarily confirming the potential of L. pontis, L. amylovorus, and L. ultunensis to convert FB1 into the less toxic metabolite HFB1, and proposing a possible adsorption-degradation mechanism. The revised version shows significant improvements in language standardization, data presentation, and discussion depth. However, key issues remain, including insufficient sample representativeness, lack of positive controls, and insufficient experimental validation of the mechanism. Therefore, I recommend that the manuscript would be more suitable for final acceptance if the authors can supplement additional experiments, such as pure culture degradation validation or heat-inactivated bacterial adsorption assays, and explicitly address these limitations in the Discussion section.

Author Response

Dear Editor,

We sincerely thank you and the reviewer for the feedback on our manuscript. We have carefully considered all suggestions, and most have been incorporated into the revised manuscript to enhance its clarity, precision, and scientific rigor. Below, we provide detailed responses to the reviewer’s comments.

Response to Reviewer 1

The reviewer expressed concern regarding the use of caecal chyme from a single animal, suggesting that pooling samples from multiple animals could reduce bias arising from individual variation. We maintain that using a single animal was sufficient for this exploratory in vitro study, as the pig was maintained under strictly controlled conditions, including uniform feed, age, and housing environment, ensuring a representative and consistent caecal microbiome. In addition, the use of one animal was appropriate for this model, as the microbiome was typical and the study did not aim to evaluate population-level microbiome variation in response to FB1 inoculation. Rather, the study was designed to evaluate the conversion of FB1 to its hydrolysed form HFB1 by the caecal microbiome, while also monitoring microbial population responses to FB1 exposure. This design was therefore suitable for assessing both FB₁ hydrolysis and the microbial response under controlled in vitro conditions. To address this concern, we have added a clarifying statement in the Methods section of the revised manuscript to explain the rationale for using a single animal, emphasizing the controlled conditions that minimized biological variability (See L278-284).

The reviewer also sought clarification regarding the positive controls, specifically whether they referred to the toxin control (C2) or the microbiota-only control (C1). We confirm that both C1 (microbiota-only control) and C2 (toxin-only control) were appropriately included to assess baseline microbial activity and toxin effects, respectively. These have been clearly described in the Materials and Methods section to explicitly delineate the roles of C1 and C2 in the experimental design (See L301-305).

The reviewer further noted that validation of the proposed FB₁ detoxification mechanism was beyond the scope of this study. We agree and have clarified this in the revised aims and objectives of the manuscript. Additionally, the reviewer suggested incorporating heat-inactivated bacterial adsorption assays to distinguish between enzymatic degradation and physical adsorption. We have addressed this by including a statement in the Discussion as well as conclusion section indicating that future studies will incorporate such adsorption assays to further elucidate the underlying detoxification mechanisms.

Thank you for the attention given to our manuscript.

Sincerely,

Dr. Isaac Malgwi (On behalf of all authors).

Round 3

Reviewer 1 Report

Comments and Suggestions for Authors

The problem has been solved.

Author Response

Dear Academic Editor, 

We would like to thank you for the significant attention our manuscript has received. We agree with your observation that the variation is reduced based on using caecal chyme from a single animal. Thank you for highlighting this question. To inform readers about the study limitation, this concern (including a few others) has been addressed in the limitation and conclusion parts of the manuscript. Notwithstanding, the present study was designed to evaluate the conversion of FB1 to its hydrolyzed form, HFB1, by the caecal microbiome. Therefore, using a caecal sample from a single animal was appropriate for this model, as the microbiome was considered representative. However, in future studies, population-level variation in the microbiome’s response to FB1 inoculation could be explored. 

Thank you.

Sincerely,

Dr. Isaac Malgwi (On behalf of all authors).